# PERTURBATION DEFOCUSING FOR ADVERSARIAL DEFENSE

## ABSTRACT

Recent research indicates adversarial attacks are likely to deceive neural systems, including large-scale, pre-trained language models. Given a natural sentence, an attacker replaces a subset of words to fool objective models. To defend against adversarial attacks, existing works aim to reconstruct the adversarial examples. However, these methods show limited defense performance on the adversarial examples whilst also damaging the clean performance on natural examples. To achieve better defense performance, our finding indicates that the reconstruction of adversarial examples is not necessary. More specifically, we inject non-toxic perturbations into adversarial examples, which can disable almost all malicious perturbations. In order to minimize performance sacrifice, we employ an adversarial example detector to distinguish and repair detected adversarial examples, which alleviates the mis-defense on natural examples. Our experimental results on three datasets, two objective models and a variety of adversarial attacks show that the proposed method successfully repairs up to $\sim 97\%$ correctly identified adversarial examples with $\leq \sim 2\%$ performance sacrifice. We provide an anonymous demonstration[1] of adversarial detection and repair based on our work.

## 1 INTRODUCTION

Neural networks have been employed achieved state-of-the-art performance on various tasks. However, recent research has shown their vulnerability to adversarial attacks (Szegedy et al., 2014; Goodfellow et al., 2015). In particular, language models have shown to be vulnerable to adversarial examples (a.k.a., adversary) (Garg & Ramakrishnan, 2020; Li et al., 2020; Jin et al., 2020; Li et al., 2021a) generated by replaced specific words in a sentence. Compared to adversarial robustness on computer vision tasks (Alzantot et al., 2018; Ren et al., 2019; Zang et al., 2020; Zhang et al., 2021; Jin et al., 2020; Garg & Ramakrishnan, 2020; Li et al., 2021a; Wang et al., 2022), text adversarial defense (a.k.a. adversarial repair) has attracted less attention resulting in limited progress in adversary defense. Moreover, the crux of adversarial defense, i.e., performance sacrifice, has not been settled by existing studies.

While the prominent works tend to solve adversarial defense via adversarial training or feature reconstruction, we propose perturbation defocusing to address adversarial defense in natural language processing. More specifically, perturbation defocusing attempts to apply non-toxic perturbations to adversaries to repair them. Although it doesn't seem to be an intuitive thought, it is motivated by empirical observations that malicious perturbations rarely destroy the fundamental semantics of a natural example. In other words, these adversaries can be easily repaired by distracting the objective model from malicious perturbations. We validate a simple implementation of perturbation defocusing with preliminary experiments: simply masking the malicious perturbations, as in Figure 1. The experimental results in Table 1 show that masking malicious perturbations repairs a considerable number of adversaries (achieves up to $91.05\%$ restored accuracy on the `Amazon Polarity` dataset). Unfortunately, the positions of malicious perturbations are unknown in real adversarial defense. We employ adversarial attackers to perform perturbation defocusing as an alternative. If an adversary is identified, we obtain its perturbed prediction and keep attacking this adversary until the new prediction differs from the former. In this way, the malicious perturbations

---

[1] https://huggingface.co/spaces/anonymous8/RPD-Demo

| Original: | This is among the year's most intriguing **explorations** of alientation. | 🙂 |
| Adversary: | This is among the year's most intriguing **scrutinize** of alientation. | 🙁 |
| Defocusing: | This is among the year's most intriguing **[MASK]** of alientation. | 🙂 |

Figure 1: A real example of perturbation defocusing, which masks the perturbed words to repair an adversary. "[MASK]" denotes the mask token. This virtual adversary is generated by TEXTFOOLER.

are defocused without knowing the positions of malicious perturbations. Because adversarial attackers have large search spaces of non-toxic perturbations, almost all malicious perturbations in adversaries can be defocused in our experiments. However, there is a prerequisite that the adversaries must be precisely identified to prevent oriented attackers from attacking natural examples (Bao et al., 2021) in perturbation defocusing. Hopefully, although existing adversarial attackers emphasize the naturalness of adversaries (Zang et al., 2020; Li et al., 2021b; Le et al., 2022), our study suggests that PLM-based models can efficiently distinguish the adversaries (refer to Figure 4), provided that the adversarial detection objective is involved in fine-tuning processing. Thereafter, we propose reactive perturbation defocusing (RPD) based on perturbation defocusing and adversary detection that alleviates performance sacrifice by only repairing detected adversaries.

We deploy RPD on a PLM-based model, and it can be extended to other NLP models. We evaluate RPD on three text classification datasets under challenging adversarial attackers. The experimental results demonstrated that RPD is capable of repairing $\sim 97\%+$ of identified adversaries without observable performance sacrifice (under $\sim 2\%$) on clean data (please refer to Table 6). In summary, our contributions are mainly as follows:

a) We propose perturbation defocusing to supersede feature reconstruction-based methods for adversarial defense, which almost repairs all correctly identified adversaries.
b) We integrate an adversarial detector with a PLM-based classification model. Based on multi-attack adversary sampling, the adversarial detector can efficiently detect most of the adversaries.
c) We evaluate RPD on multiple datasets, PLMs and adversarial attackers. The experimental results indicate that RPD has an impressive capacity to detect and repair adversaries without sacrificing clean performance.

## 2 RELATED WORKS

Existing adversarial defense studies can be coarsely classified into three types: adversarial training-based approaches (Miyato et al., 2017; Zhu et al., 2020; Ivgi & Berant, 2021); context reconstruction-based methods (Pruthi et al., 2019; Liu et al., 2020b; Mozes et al., 2021; Keller et al., 2021; Chen et al., 2021; Xu et al., 2022; Li et al., 2022; Swenor & Kalita, 2022); and feature reconstruction-based methods(Zhou et al., 2019; Jones et al., 2020; Wang et al., 2021a). In the meantime, some research(Wang et al., 2021b) explores hybrid defenses against adversarial attacks. Neverthe-

Table 1: The experimental performance of masking-based perturbation defocusing on adversaries.

| Dataset | Clean Acc.(%) | Attacker | Attacked Acc.(%) | Restored Acc.(%) |
|---|---|---|---|---|
| SST2 | 92.03 | BAE | 45.96 | 59.80 |
| | | PWWS | 29.82 | 74.37 |
| | | TEXTFOOLER | 22.02 | 72.27 |
| Amazon Polarity | 96.36 | BAE | 56.65 | 78.58 |
| | | PWWS | 19.40 | 81.25 |
| | | TEXTFOOLER | 20.80 | 91.07 |
| AGNews | 91.35 | BAE | 81.80 | 71.85 |
| | | PWWS | 56.55 | 86.99 |
| | | TEXTFOOLER | 32.95 | 83.33 |

less, there are some problems that remain with the existing methods. For example, due to the issue of catastrophic forgetting (Dong et al., 2021), adversarial training has been shown to be inadequate for improving the robustness of PLMs in fine-tuning. On the contrary, it significantly increases the cost of objective model training. For context reconstruction (e.g., word substitution and translation-based reconstruction), these methods sometimes fail to identify semantically repaired adversaries or have a tendency to introduce new malicious perturbations (Swenor & Kalita, 2022). In recent studies, it has been recognised that feature (e.g., embedding) space reconstruction-based approaches are more successful than context reconstruction methods like word substitution (Mozes et al., 2021;

Bao et al., 2021). However, these feature reconstruction methods may have difficulty repairing typo attacks (Liu et al., 2020a; Tan et al., 2020; Jones et al., 2020), sentence-level attacks(Zhao et al., 2018; Cheng et al., 2019), and other unknown attacks. These studies usually limit the experiments to word substitution-based attacks (typically Genetic Algorithm (Alzantot et al., 2018)). In contrast to prior efforts, we argue that reconstruction is not necessary for adversarial repair. Because the fundamental semantics of an adversary generally remains in the adversary, we just need to distract objective models' attention from malicious perturbations. Another problem with the existing methods is that they neglect the importance of adversary detection and assume that all instances are adversaries, resulting in numerous unsuccessful defenses. Compared to existing works, our study focuses on reactive adversarial defense and addresses the crux of performance sacrifice brought on by adversarial defense.

# 3 METHOD

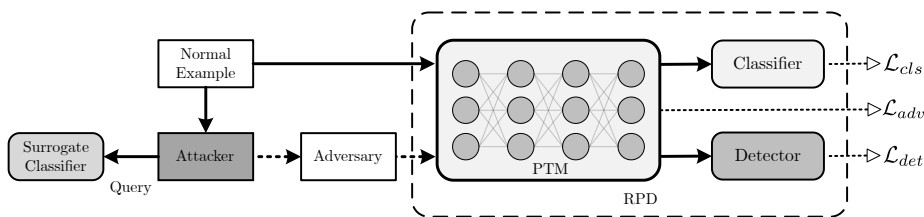

Phase #1: Multi-objective Fine-tuning

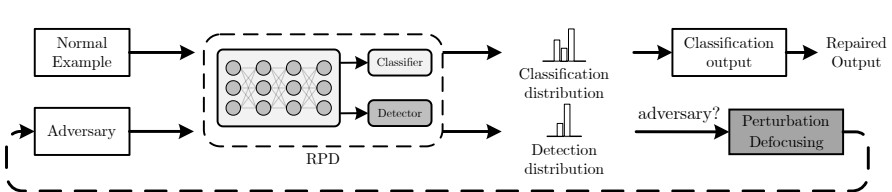

Phase #2: Adversary Detection & Repair

Figure 2: The framework of RPD. The dotted lines with solid arrow means the steps depends on the existence of an adversary, while the dotted lines with triangles denotes the objectives for multitask training. In addition to standard classification objective, RPD contains an adversary detection objective and a detached adversarial training objective.

We illustrate the framework of RPD in Figure 2, which consists two phases: multi-objective fine-tuning and adversarial repair. In Phase #1, we fine-tune RPD based on three training objectives, including the original classification objective. Next, we introduce each objective in following sections.

## 3.1 ADVERSARY DETECTOR TRAINING

Since we train the adversary detector using supervise learning, we will introduce how to sample adversaries by adversarial attackers.

### 3.1.1 TEXT ADVERSARIAL ATTACK

We focus on word-level adversarial attacks in this work. Let $x = (w_1, w_2, \cdots, w_n)$ be a natural sentence, where $w_i$, $1 \leq i \leq n$, denotes a word; $y$ is the ground truth label. The word-level attackers replace the original words with their words (e.g., synonyms) to fool the objective model. For example, substituting $w_i$ with $\hat{w}_i$ will generate an adversary: $\hat{x} = (w_1, \hat{w}_2, \cdots, w_n)$, where $\hat{w}_i$ is a alternative of $w_i$. The objective model $F$ predict $\hat{x}$ as follows:

$$\hat{y} = \arg\max F\left(\cdot | \hat{x}\right), \tag{1}$$

where $\hat{y} \neq y$ if $\hat{x}$ is a successful adversaries. The perturbations in $\hat{x}$ are expected to be human-imperceptible. However, most of existing attackers tend to introduce grammatical, syntactical errors to a certain extent, while the features of these errors in $\hat{x}$ can be easily modeled by a PLM.

### 3.1.2 Multi-attack Adversary Sampling

Based on the open-source adversarial attack methods (i.e., attacker), we perform multi-attack sampling (line $2 - 12$ in Algorithm 1) to train the adversary detector. Let $\mathcal{D}_{nat}$ be the natural examples. $\forall x \in \mathcal{D}_{nat}$, we try to find a successful adversary as follows:

$$\hat{x}, \hat{y} \leftarrow \sum_{i=1}^{k} \mathcal{A}_i(F_s, x, y), \tag{2}$$

where $\leftarrow$ indicates the adversary search process; $\hat{x}$, $\hat{y}$ are the perturbed sentence and label. Note that if the attack failed, $\hat{y} = y$ but $\hat{x} \neq x$. $\mathcal{A}_i$, $1 \leq i \leq k$, is a attacker for adversary sampling; $k$ is the number of sampling attackers. $F_s$ is the surrogate classifier trained on natural examples (line 6 in Algorithm 1). The label $\tilde{y}$ of an example in RPD contains 3 sub-labels (for the objectives of classification ($\mathcal{L}_{cls}$), detached adversarial training($\mathcal{L}_{adv}$), adversarial detection($\mathcal{L}_{det}$), respectively). In the sampling process, $\tilde{y}$ is conditioned on the attack result (lines $6 - 9$ in Algorithm 1):

$$\tilde{y} := \begin{cases} (\phi, y, 0), & \hat{y} = y \\ (y, \phi, 1), & \hat{y} \neq y \end{cases}, \tag{3}$$

where $\phi$ indicates the sub-label is neglected in cross-entropy loss calculation. All adversaries and natural examples are fused to train an adversary detector. We also conduct experiments on single-attack sampling-based RPD (denoted as S-RPD) to evaluate the significance of multi-attack sampling (please refer to Table 5).

### 3.1.3 Adversary Detector Objective

After adversary sampling, we fit the adversary detector[2] on natural examples and sampled adversaries. Let $\mathbf{H}$ be the representation of an example encoded by a PLM, RPD calculate the adversarial distribution as follows:

$$\hat{\iota}_i = \frac{\exp\left(pool(\mathbf{H})_i\right)}{\sum_{j=1}^{2} \exp\left(pool(\mathbf{H})_j\right)}, \tag{4}$$

where $\hat{\iota}_i$, $1 \leq i \leq 2$, indicates whether a sentence has been perturbed; $pool$ is the head pooling of PLM. The adversarial detection objective can be formulated as:

$$\mathcal{L}_{det} = -\sum_{i=1}^{2} \hat{\iota}_i \log \iota_i, \tag{5}$$

where $\iota_i$ denotes the true adversarial label. Because the adversary detector is a binary text classifier, we adopt widely used cross-entropy to minimize $\mathcal{L}_{det}$.

### 3.2 Detached Adversarial Training

We employ adversarial training in RPD as it has been recognized to be able to improve robustness (Miyato et al., 2017; Zhu et al., 2020; Ivgi & Berant, 2021). However, we find that traditional adversarial training may degenerate performance on natural examples. Hence, we propose the detached adversarial training objective to simultaneously mitigate performance sacrifice and improve objective model's robustness. The detached adversarial training objective $\mathcal{L}_{adv}$ can be formulated as:

$$\min \mathbb{E}_{(x,y)\sim\mathcal{D}_{nat}} \left[ \max_{\hat{x}, \hat{y} \leftarrow \mathcal{A}(x,y)} \mathcal{L}_{adv}(\hat{x}, y) \right]. \tag{6}$$

More specifically, the standard classifier only learns to classify natural examples, while the adversarial training objective only involves the adversaries. To clarify each step, we describe the training of RPD in Algorithm 1. The efficacy analysis of the detached adversarial training objective is available in Table 9.

---

[2]Generally, an independent adversarial detection method also works in RPD, but the PLM-based adversary detector is simple and efficient.

### 3.3 STANDARD CLASSIFICATION TRAINING

The last objective $\mathcal{L}_{cls}$ aims at standard classification. We employ cross-entropy to optimize the standard classifier as following:

$$\mathcal{L}_{cls} = -\sum_{1}^{C} \hat{y}_i \log y_i, \tag{7}$$

$$\mathcal{L}_{rpd} := \mathcal{L}_{cls} + \alpha \mathcal{L}_{det} + \beta \mathcal{L}_{adv} + \lambda \|\Theta\|_2, \tag{8}$$

where $\hat{y}_i$, $1 \leq i \leq C$, is the prediction of classification; $C$ indicates the classes number. $\mathcal{L}_{rpd}$ is the overall objective of RPD. $\alpha$ and $\beta$ are the objective weights. In this work, $\alpha$ and $\beta$ are set to 5 by grid searching. $\lambda = 10^{-5}$ is the $L_2$ regularization parameter; $\Theta$ denotes the parameter set of RPD.

---

**Algorithm 1:** Adversarial sampling and training of RPD

---

**Require:** $\mathcal{D}_{nat}$, attackers $\{\mathcal{A}\}_{i=1}^{k}$
**Output :** RPD model $F_R$ for adversary detection
1  Train a surrogate classifier $F_s$ on $\mathcal{D}_{nat}$ for adversaries sampling;
2  $\mathcal{B} \leftarrow \emptyset$;
3  **for** $i \leftarrow 1$ **to** $k$ **do**
4     **forall** $(x, y) \in \mathcal{D}_{nat}$ **do**
5        $\hat{x}, \hat{y} \leftarrow \mathcal{A}_i(F_s, x, y)$;
6        **if** $\hat{y} \neq y$ **then**
7           $\mathcal{B} := \mathcal{B} \bigcup \{(\hat{x}, (\phi, y, 1))\}$;
8        **end**
9        $\mathcal{B} := \mathcal{B} \bigcup \{(x, (y, \phi, 0))\}$;
10    **end**
11 **end**
12 Train $F_R$ on $\mathcal{B}$ using $\mathcal{L}_{rpd}$;
13 **return** $F_R$

---

**Algorithm 2:** Adversarial detection and defense based on RPD

---

**Input  :** Input examples $\mathcal{D}_e$; attacker $\mathcal{A}_{PD}$ for perturbation defocusing
**Output:** The Repaired Outputs $\mathcal{R}$
1  $\mathcal{R} \leftarrow \emptyset$;
2  **forall** $x_e \in \mathcal{D}_e$ **do**
3     $\hat{y}, \hat{\imath} = F_R(x_e)$;
4     **if** $\hat{\imath} == 1$ **then**
5        $x_r \leftarrow \mathcal{A}_{PD}(x_e, \hat{y})$;
6        $\hat{y}_r, \hat{\imath}_r = F_R(x_r)$;
7        $\mathcal{R} := \mathcal{R} \bigcup \{\hat{y}_r\}$;
8     **end**
9     **else**
10       $\mathcal{R} := \mathcal{R} \bigcup \{\hat{y}\}$;
11    **end**
12 **end**
13 **return** $\mathcal{R}$

---

### 3.4 REACTIVE PERTURBATION DEFOCUSING

In the Phase #2, RPD tries to repair the identified adversaries via perturbation defocusing (Algorithm 2). Assuming that the $\hat{x}$, $\hat{\imath} \leftarrow F_R(\tilde{x})$ denote the classification distribution and adversarial detection distribution from RPD. If $\hat{\imath}$ (i.e., $\hat{\imath}$ is 1) indicates an adversary, the repaired example $x_r$ is derived by:

$$x_r \leftarrow \mathcal{A}_{PD}(\tilde{x}, \hat{y}), \tag{9}$$

where $\mathcal{A}_{PD}$ is an adversarial attacker performing perturbation defocusing. Finally, the repaired adversaries's output $\hat{y}_r \leftarrow F_R(x_r)$ (lines $6 - 7$ in Algorithm 2). Note that the adversaries repaired by perturbation defocusing are still perturbed examples, but no more perturbation defocusing is needed for repaired adversaries.

## 4 EXPERIMENTS

### 4.1 DATASETS AND EVALUTAION METRICS

To validate the efficacy of RPD, we conduct experiments on three classification datasets [3]: SST2[4], Amazon Polarity[5] and AGNews[6] datasets, respectively. SST2 and Amazon

---

[3]Note that attacking the PLM-based models is very expensive. In this case, we use the subsets of Amazon Polarityand AGNews datasets in our experiments, the numbers of examples in the these subsets are $10K$. We submit the datasets as supplementary materials for reproducible evaluation.

[4]https://huggingface.co/datasets/sst2

[5]https://huggingface.co/datasets/amazon_polarity

[6]https://huggingface.co/datasets/ag_news

| Dataset | Categories | Number of Examples | | | |
|---|---|---|---|---|---|
| | | Training Set | Validation Set | Testing Set | Sum |
| SST2 | 2 | 6920 | 872 | 1821 | 9613 |
| Amazon Polarity | 2 | 8000 | 0 | 2000 | 10000 |
| AGNews | 4 | 7000 | 1000 | 2000 | 10000 |

Table 2: The details of experimental datasets used for evaluating RPD. We further split the original training set into training and validation subsets for the AGNews dataset.

Polarity datasets are binary sentiment classification datasets, while AGNews is a news classification dataset containing 4 classes. Table 2 shows the dataset details. For detailed evaluation metric clarification, please refer to Appendix A.1.3.

## 4.2 EXPERIMENTS SETTING

The adversarial defense experiments involve attack methods and PLM[7]-based classifiers. We adopt the open-source implementations of the adversarial attack methods from TextAttack[8] as candidate attackers, following the original attack settings. We use BERT and DEBERTA as objective classifiers to evaluate the adversarial repair performance, while DEBERTA is the base objective model used in all ablation experiments. In Table 3, we evaluate adversarial detection and defense performance across the whole testing set. However, we only evaluate 500 examples in the research questions due to resource limitation. For detailed hyper-parameter settings, please refer to A.1.1.

## 4.3 ADVERSARIAL ATTACKERS

The attacker for perturbation defocusing is PWWS in this work, because it hardly corrupts the semantics in the repaired adversaries compared to BAE and is slightly faster than TEXTFOOLER. The attackers used for adversarial sampling are BAE, PWWS and TEXTFOOLER. We briefly introduce these attackers as follows:

**PWWS** (Ren et al., 2019) is a synonym-substitution based adversarial attack method. PWWS combines both the word saliency and the classification probability to perform word replacement.

**BAE** (Garg & Ramakrishnan, 2020) replaces and inserts tokens according to alternatives generated by a masked language model (MLM). To identify the essential words, BAE employs a deletion-based measure of word significance.

**TEXTFOOLER** (Jin et al., 2020) takes more constraints (e.g., prediction consistency, semantic similarity and fluency) into consideration in generating adversaries. TEXTFOOLER adopts a gradient-based word importance measure to locate and perturb the important words.

The other attackers used in ablation experiments are: PSO (Zang et al., 2020), IGA (Wang et al., 2021a), DEEPWORDBUG (Gao et al., 2018), CLARE (Li et al., 2021a).

## 4.4 COMPARED METHODS

**RPD**: The baseline of RPD that adopts multi-attack sampling based on BAE, PWWS and TEXTFOOLER. The main experimental results of RPD are listed in Table 3.

**S-RPD**: The variant of RPD that samples adversaries from a targeted single attack. We evaluate the transferability of S-RPD and show the results in Table 4 and Table 8.

**RAT**: RAT has an adversarial classifier based on reactive adversarial training. RAT predicts adversaries using an adversarial classifier and predicts natural examples using a standard classifier. The number of adversaries used in training RAT is the same as the number of RPD's training examples.

We also compare the adversarial defense performance of RPD with other state-of-the-art methods, such as ASCC and RIFT. Please refer to Appendix A.3 for more details.

---

[7]We use transformers to implement RPD: https://github.com/huggingface/transformers
[8]https://github.com/QData/TextAttack

## 4.5 Main Results

Table 3: The adversarial detection and defense performance of RPD on different objective models; "Acc." is an abbreviation for Accuracy. The results are the medians in five runs.

| Dataset | Target Model | Clean Acc.(%) | Attacker | Attacked Acc.(%) | Defender | Detection Acc.(%) | Defense Acc.(%) | Restored Acc.(%) |
|---|---|---|---|---|---|---|---|---|
| SST2 | BERT | 91.76 | BAE | 38.93 | RPD | 66.86 | 65.02 | 70.40 |
| | | | PWWS | 14.44 | | 90.50 | 90.50 | 83.14 |
| | | | TextFooler | 6.21 | | 90.81 | 89.90 | 81.82 |
| | DeBERTA | 94.73 | BAE | 45.96 | | 70.55 | 69.07 | 77.70 |
| | | | PWWS | 29.82 | | 95.09 | 94.86 | 91.21 |
| | | | TextFooler | 22.02 | | 94.33 | 92.74 | 89.13 |
| | | | BAE | 45.96 | RAT | 67.55 | 12.05 | 43.66 |
| | | | PWWS | 29.82 | | 94.63 | 15.87 | 34.65 |
| | | | TextFooler | 22.02 | | 91.76 | 19.61 | 28.17 |
| Amazon Polarity | BERT | 94.55 | BAE | 44.00 | RPD | 79.15 | 79.15 | 88.60 |
| | | | PWWS | 4.10 | | 95.64 | 95.64 | 91.85 |
| | | | TextFooler | 1.25 | | 94.94 | 94.94 | 91.60 |
| | DeBERTA | 96.20 | BAE | 56.65 | | 86.22 | 86.22 | 91.55 |
| | | | PWWS | 19.40 | | 96.25 | 96.25 | 94.65 |
| | | | TextFooler | 20.80 | | 95.66 | 95.66 | 92.65 |
| | | | BAE | 56.65 | RAT | 89.39 | 33.61 | 75.90 |
| | | | PWWS | 19.40 | | 96.57 | 29.60 | 48.20 |
| | | | TextFooler | 20.80 | | 95.97 | 38.43 | 53.20 |
| AGNews | BERT | 91.50 | BAE | 74.80 | RPD | 43.82 | 43.07 | 83.95 |
| | | | PWWS | 28.55 | | 91.67 | 89.34 | 87.65 |
| | | | TextFooler | 10.50 | | 89.63 | 87.01 | 84.10 |
| | DeBERTA | 92.12 | BAE | 81.80 | | 87.66 | 85.15 | 89.15 |
| | | | PWWS | 56.55 | | 97.30 | 95.89 | 90.95 |
| | | | TextFooler | 32.95 | | 93.27 | 91.37 | 88.40 |
| | | | BAE | 81.80 | RAT | 88.68 | 20.71 | 73.75 |
| | | | PWWS | 56.55 | | 97.29 | 18.75 | 58.25 |
| | | | TextFooler | 32.95 | | 90.86 | 34.33 | 59.80 |

The experimental findings in Table 3 show how well RPD is able to identify and defend against adversaries. We provide both the standard classification performance and the accuracy under adversarial attack of the objective models in order to intuitively demonstrate the efficacy of adversarial detection and repair. As demonstrated in existing studies (Jin et al., 2020; Garg & Ramakrishnan, 2020), the objective models' performance is generally significantly decreased by adversarial attackers, particularly on the SST2 and Amazon Polarity datasets. For example, BERT's performance can be decreased by up to $90\%+$, and its accuracy on the Amazon Polarity dataset is only $1.25\%$ at its worst(TextFooler). In general, DeBERTA is more robust than BERT in the majority of circumstances; its worst accuracy on Amazon Polarity dataset is $19.4\%$(under PWWS attack). In a nutshell, adversarial attacks continue to be a threat to existing PLMs. Despite having more classes, AGNews only sacrifices $11.32\%$ and $16.7\%$ accuracy when attacked by BAE, which means the PLM's robustness varies depending on the dataset domain.

Overall, RPD's ability in terms of adversarial detection and repair is encouraging. Among all the datasets, RPD based on multi-attack sampling performs impressively, demonstrating that PLMs (especially DeBERTA) are capable of recognising adversaries. Meanwhile, compared to previous adversarial defense studies, the regression of standard classification and adversarial detection error rate on natural examples are as low as $\sim 1\%$ and $\sim 10\%$, respectively (please refer to Appendix A.2 for details). This reduces mis-repairs on natural examples. The adversarial defense performance based on perturbation defocusing depends on the accuracy of adversarial detection, which means detection accuracy $\geq$ defense accuracy. However, because the accuracy on natural examples suffers no significant loss, in the case of the worst detection accuracy ($43.82\%$ on AGNews dataset) of BERT, the restored accuracy ($83.95\%$) is still better than BERT without defense ($74.8\%$). On the one hand, our experimental results show reactive perturbation defocusing is able to repair $\sim 97\%+$ of correctly identified adversaries without clean performance sacrifice. On the other hand, RPD can be adapted to other models provided that the adversary detectors are deployed.

To our best knowledge, there is no reactive defense counterpart that can be directly compared with RPD. Hence, we implement RAT based on reactive adversarial training. It can be observed that reactive adversarial training has worse performance of adversarial repair comapred to RPD, with $\sim\leq 30\%$ defense accuracy and $\sim\leq 60\%$ restored accuracy in most situations. This means that RAT can hardly handle challenging adversarial attacks, especially while the number of adversaries is limited. Due to resource limitations, we show a part of the experimental results compared with other popular proactive adversarial defense methods in Appendix A.3.

### 4.6 RESEARCH QUESTIONS

We discuss more findings about RPD by answering the following research questions.

### RQ1: DOES PERTURBATION DEFOCUSING REALLY REPAIR ADVERSARIES?

Our main experimental results show that perturbation defocusing is able to repair $97\%+$ of correctly discriminated adversaries. To explain why PD works, we investigate the similarity between adversaries and repaired adversaries. We randomly select $500$ natural examples from SST2, Amazon Polarity and AGNews datasets and obtain the adversaries and repaired adversaries. We encode these examples and calculate the output cosine similarity between adversary-natural example pairs and repaired adversary-natural example pairs. We plot the cumulative distributions of similarity scores on the SST2 dataset in Figure 3 (the visualizations of other datasets are available in Figure 5). The cumulative distributions on the SST2 dataset show the repaired adversaries resemble natural examples from the perspective of predictions ($\Delta_{rep} \geq 0.92$), while the adversaries generally have $\Delta_{rep} \leq -0.85$. This indicates the adversaries repaired by perturbation defocusing contain similar semantics to natural examples.

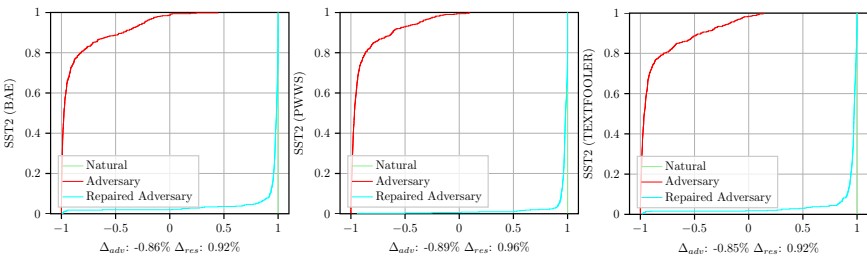

Figure 3: The cumulative distribution of output's cosine similarity scores towards natural examples. $\Delta_{adv}$ and $\Delta_{rep}$ indicate the average similarity scores of adversaries and repaired adversaries.

We also visualize the similarity of the feature space. We encode the above examples and visualize the representations via $t$-SNE in Figure 4(the visualizations of other datasets are available in Figure 6). It can be observed that the repaired adversaries are still discriminateable by PLMs because their feature space is similar to the adversaries. However, we note that more repaired adversaries lie in the natural example space compared to adversaries, which means repaired adversaries are more similar to natural examples in the feature space to some extent.

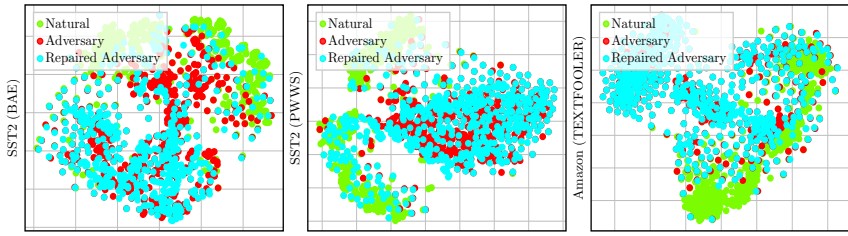

Figure 4: The cluster visualizations of natural examples, adversaries and repaired adversaries via $t$-SNE.

### RQ2: CAN RPD WORKS ON UNKNOWN ADVERSARIAL ATTACKS?

The most challenging obstacle for adversarial detection and repair methods is working on unknown attacks. Because RPD relies on a simple PLM-based adversarial detector to identify adversaries, we need to know whether it can distinguish adversaries generated by unknown adversarial attackers. In this case, we evaluate the RPD's performance on unknown attacks. The results are available in Table 4(we also evaluate the transferability of S-RPD in Table 8). The experimental results in Table 4 show that even though trained on BAE, PWWS and TEXTFOOLER, RPD is able to distinguish

unknown adversaries, especially for `PSO` and `DEEPWORDBUG`. For example, the accuracy of repaired adversaries is promising ($97.5\%$ and $90.48\%$ on `SST2` and `Amazon Polarity` datasets). However, there is a significant defense performance drop in the adversaries generated by `CLARE`. In conclusion, `RPD` can identify and repair unknown adversaries.

Table 4: The adversarial detection and defense performance of `RPD` on unknown attacks.

| Dataset | Clean Acc.(%) | Attacker | Attacked Acc.(%) | Detection Acc.(%) | Defense Acc.(%) | Restored Acc.(%) |
|---|---|---|---|---|---|---|
| SST2 | 94.73 | PSO | 7.95 | 87.50 | 87.50 | 82.61 |
| | | IGA | 7.52 | 92.11 | 88.34 | 73.33 |
| | | DEEPWORDBUG | 22.22 | 98.44 | 87.50 | 90.00 |
| | | CLARE | 1.39 | 62.50 | 59.37 | 57.00 |
| Amazon Polarity | 96.20 | PSO | 5.76 | 90.48 | 90.48 | 91.55 |
| | | IGA | 14.91 | 92.31 | 92.31 | 94.65 |
| | | DEEPWORDBUG | 43.43 | 87.04 | 85.19 | 86.87 |
| | | CLARE | 3.25 | 58.82 | 58.82 | 53.33 |
| AGNews | 92.12 | PSO | 12.07 | 63.46 | 59.62 | 89.15 |
| | | IGA | 27.51 | 40.74 | 40.74 | 90.95 |
| | | DEEPWORDBUG | 45.00 | 92.73 | 89.09 | 85.00 |
| | | CLARE | 8.46 | 61.54 | 61.54 | 50.00 |

### RQ3: DOES MULTI-ATTACK SAMPLING OUTPERFORM SINGLE-ATTACK SAMPLING?

We find that multi-attack sampling may assist adversarial detectors in differentiating between hostile cases. In order to verify our idea, we perform ablation experiments based on single-attack sampling (i.e., `S-RPD`) and provide the results in Table 5. In the majority of instances, the detection accuracy of `S-RPD` suffers large decreases (up to $12.6\%$). Consequently, the repair performance demonstrates up to $18.21\%$ regression. We attribute the degraded performance of adversarial detection to two factors: a) single-attack sampling leads to fewer training data for the adversarial detector; b) multi-attack sampling may generate more diverse adversarial patterns than single-attack sampling. In summary, defense accuracy and restored accuracy show that single-attack sampling limits `RPD`'s performance.

Table 5: The adversarial detection and defense performance of `S-RPD` under different attackers and PLMs. The "`Diff`" measures the performance change compared to `RPD`.

| Dataset | Target Model | Clean Acc.(%) | Attacker | Attacked Acc.(%) | Detection Acc.(%) | Detection Diff.(%) | Defense Acc.(%) | Defense Diff.(%) | Restored Acc.(%) | Restored Diff.(%) |
|---|---|---|---|---|---|---|---|---|---|---|
| SST2 | BERT | 91.76 | BAE | 38.93 | 55.10 | −11.76 | 53.53 | −11.49 | 64.91 | −5.49 |
| | | | PWWS | 14.44 | 79.08 | −11.42 | 78.95 | −11.55 | 74.63 | −8.51 |
| | | | TEXTFOOLER | 6.21 | 80.39 | −10.42 | 78.80 | −11.10 | 74.08 | −7.74 |
| | DEBERTA | 94.73 | BAE | 45.96 | 61.42 | −9.13 | 59.94 | −9.13 | 72.87 | −4.83 |
| | | | PWWS | 29.82 | 86.59 | −8.50 | 86.45 | −8.41 | 84.68 | −6.53 |
| | | | TEXTFOOLER | 22.02 | 87.59 | −6.74 | 85.42 | −7.32 | 85.67 | −3.46 |
| Amazon Polarity | BERT | 94.55 | BAE | 44.00 | 66.09 | −13.06 | 66.09 | −13.06 | 79.80 | −8.80 |
| | | | PWWS | 4.10 | 92.45 | −3.19 | 92.45 | −3.19 | 88.90 | −2.95 |
| | | | TEXTFOOLER | 1.25 | 87.95 | −6.99 | 87.95 | −6.99 | 84.75 | −6.85 |
| | DEBERTA | 96.20 | BAE | 56.65 | 93.56 | 7.34 | 93.56 | 7.34 | 92.75 | 1.20 |
| | | | PWWS | 19.40 | 96.72 | 0.47 | 95.33 | −0.92 | 88.80 | −5.85 |
| | | | TEXTFOOLER | 20.80 | 96.49 | 0.83 | 96.49 | 0.83 | 93.95 | 1.30 |
| AGNews | BERT | 91.50 | BAE | 74.80 | 25.61 | −18.21 | 24.28 | −18.79 | 72.90 | −11.05 |
| | | | PWWS | 28.55 | 81.76 | −9.91 | 80.96 | −8.38 | 76.65 | −11.00 |
| | | | TEXTFOOLER | 10.50 | 76.60 | −13.03 | 75.35 | −11.66 | 71.50 | −12.60 |
| | DEBERTA | 92.12 | BAE | 81.80 | 87.37 | −0.29 | 84.04 | −1.11 | 84.00 | −5.15 |
| | | | PWWS | 56.55 | 91.67 | −5.63 | 89.34 | −6.55 | 87.65 | −3.30 |
| | | | TEXTFOOLER | 32.95 | 89.63 | −3.64 | 87.01 | −4.36 | 84.10 | −4.30 |

## 5 CONCLUSION

Existing approaches for adversarial defense generally result in performance sacrifices on natural examples. In this study, we propose the `RPD` based on perturbation defocusing that alleviates performance sacrifice by only repairing identified adversaries. Perturbation defocusing exploits adversarial attacks to distract objective models from malicious perturbation and has been shown to repair up to $\sim 97\%$ of correctly identified adversaries among several challenging attackers. Perturbation defocusing is a new perspective for future adversary repair research, which may supersede the reconstruction-based methods. However, the adversarial defense performance of RPD depends on the accuracy of adversarial detection, which limits `RPD`'s performance. In the future, we will explore other adversarial detection methods and explicit constraints of semantic similarity in perturbation defocusing to improve `RPD`'s defense robustness.

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

# A  APPENDIX

## A.1  IMPLEMENTATION DETAILS

### A.1.1  HYPER-PARAMETER SETTINGS

We use the following configurations to fine-tune classifiers:

1. The learning rates for both BERT and DEBERTA are $2 \times 10^{-5}$.
2. The batch size and maximum sequence modeling length are 16 and 80, respectively.
3. The dropouts are set to 0.5 for all models.
4. The loss functions of all objectives are cross-entropy.
5. The objective models and `RPD` models are trained with 5 epochs.
6. The optimizer used for fine-tuning objective models is `AdamW`.

### A.1.2 EXPERIMENT ENVIRONMENT

The experiments are conducted on Cent OS 7, which is equipped with an RTX 3090 GPU and a Core i-12900k. We use PyTorch 1.12 and a revised version of TextAttack based on v0.3.7.

### A.1.3 METRIC CLARIFICATIONS

The clean accuracy and attacked accuracy denote the objective model's original (i.e., clean) performance and performance under attacks. The detection accuracy and defense accuracy measure the RPD's performance in adversarial detection and repair, which only measure adversaries. As a global evaluation, the restored accuracy denotes the objective model's performance on the attacked dataset (i.e., replacing the natural examples with their adversaries in the dataset if their adversaries exist.). We terminate an attack if it takes longer than 10 minutes and ignore the example in the metrics calculation.

## A.2 PERFORMANCE ON CLEAN DATA

The adversarial defense performance depends on the adversarial detection accuracy. In this case, we evaluate the adversarial detection error rate and the classification accuracy on clean data without defense. From the experimental results listed in Table 6, we observe that `RPD` achieves up to 90+ adversarial detection accuracy, which indicates if we use `RPD` as a regular classifier, the original performance will not significantly decrease. On the other hand, the classification accuracy of adversaries also benefits from the adversarial detection training objective, e.g., `SST2` and `AGNews` datasets.

Table 6: The performance of `RPD` on clean data

| Dataset | Target Model | Clean Acc.(%) | Attacker | Detection | | Classification | |
|---|---|---|---|---|---|---|---|
| | | | | Acc.(%) | Error(%) | Acc.(%) | Diff.(%) |
| SST2 | DEBERTA | 94.73 | BAE | 95.00 | 5.00 | 95.11 | 0.38 |
| | | | PWWS | 97.31 | 2.69 | 95.06 | 0.33 |
| | | | TEXTFOOLER | 97.03 | 2.97 | 95.99 | 1.26 |
| | | | Multi-attack | 90.33 | 9.67 | 94.89 | 0.16 |
| Amazon Polarity | DEBERTA | 96.20 | BAE | 95.80 | 4.20 | 95.85 | −0.35 |
| | | | PWWS | 99.01 | 0.99 | 96.92 | 0.72 |
| | | | TEXTFOOLER | 98.15 | 1.85 | 96.20 | 0.00 |
| | | | Multi-attack | 95.93 | 4.07 | 95.67 | −0.53 |
| AGNews | DEBERTA | 92.12 | BAE | 98.85 | 1.15 | 90.95 | −1.17 |
| | | | PWWS | 99.25 | 0.75 | 92.50 | 0.38 |
| | | | TEXTFOOLER | 98.40 | 1.60 | 92.45 | 0.33 |
| | | | Multi-attack | 97.60 | 2.40 | 92.70 | 0.58 |

## A.3 COMPARISON WITH ASCC

From the perspective of adversarial repair, `RPD` achieves impressive results compared with existing methods (e.g., `ASCC`). The experimental results are available in Table 7. The experimental results show that perturbation defocusing which distracts the objective model from the malicious perturbations achieves comparable performance. We explain why perturbation defocusing works for adversarial defense in Figure 3.

Table 7: The adversary defense performance comparison on `IMDB` dataset between `RPD` and other state-of-the-art defense methods under `GA` attack. * means that due to computation resource limitation, we sampled 100 adversaries generated by `GA` to train `RPD` , which is not enough (e.g., the whole training set contains 8000 examples).

| Method | Dataset | Model | Defense Acc.(%) |
|--------|---------|-------|-----------------|
| ASCC | IMDB | BERT | 70.20 |
| RIFT | IMDB | BERT | 77.20 |
| RPD* | IMDB | BERT | 81.31 |

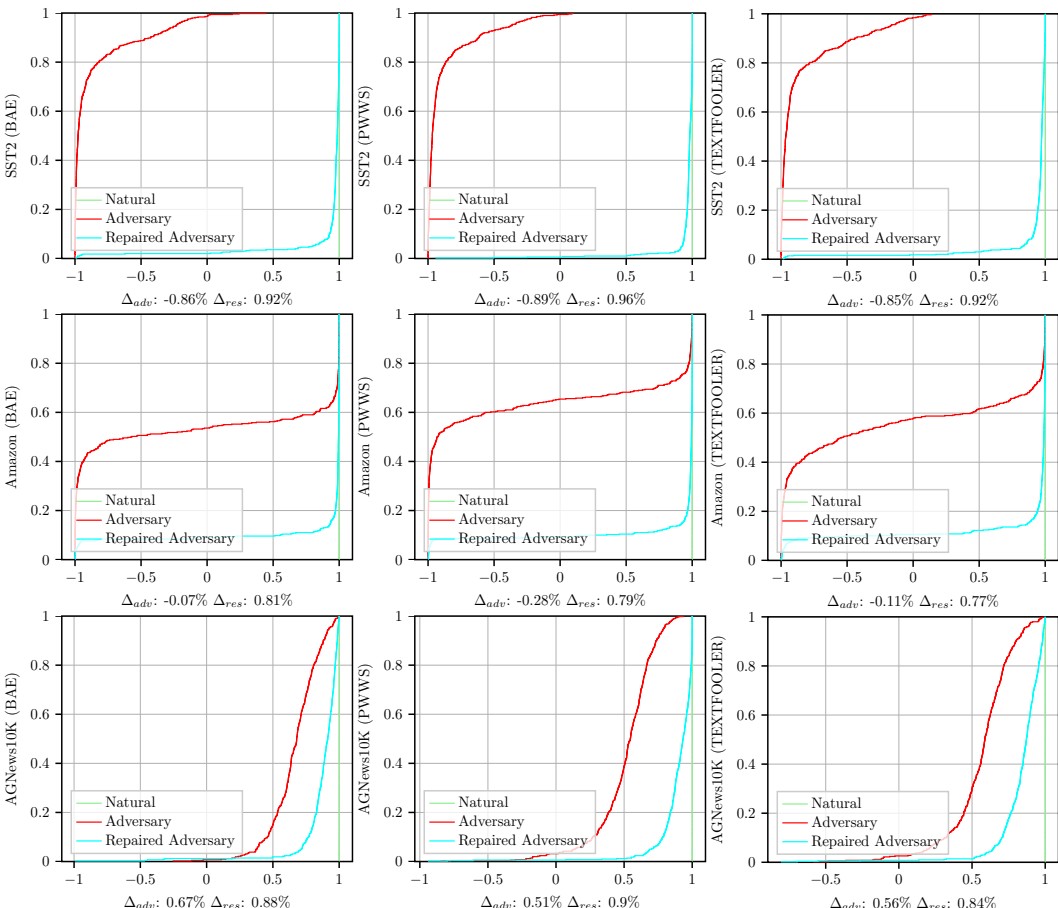

Figure 5: The cumulative distribution of output's cosine similarity scores towards natural examples. $\Delta_{adv}$ and $\Delta_{res}$ indicate the average cosine similarity scores of adversaries and defocused adversaries.

## A.4    FULL VISUALIZATIONS OF RQ1

## A.5    TRANSFER EXPERIMENTS OF S-RPD

We show the performance of `RPD` in transfer experiments in Table 8. Interestingly, the stronger the naturalness constraints cause the worse transfer ability of the adversarial detectors, e.g., `PWWS` and `TEXTFOOLER` suffer from up to 62.7% and 62.75% adversarial detection performance and adversarial repair performance drop on the `BAE`-based adversaries, especially on the `SST2` dataset. Therefore, we argue that it is imperative to train the detector to simultaneously consider attackers with different constraint strengths.

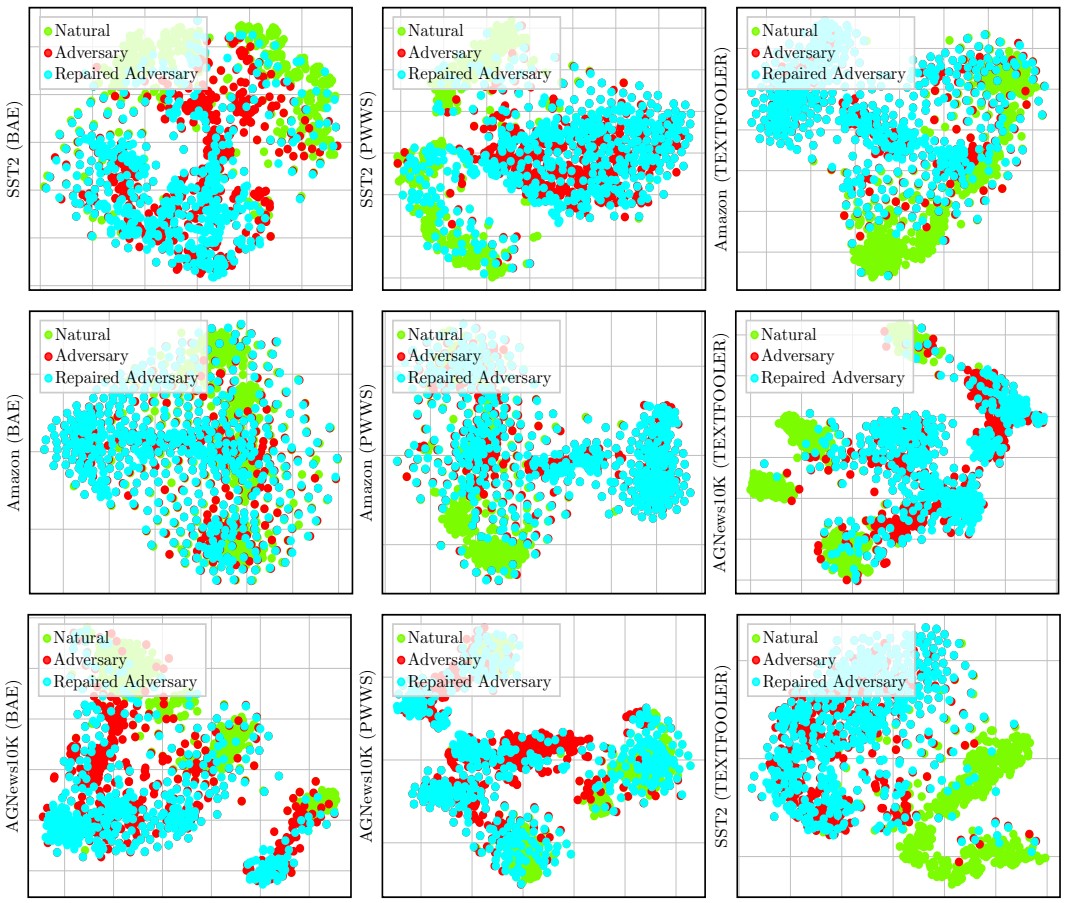

Figure 6: The $t$-SNE cluster visualizations of natural examples, adversaries and restored examples. The average cosine similarity scores of the clusters are indicated below the figures.

Table 8: The transferred performance of single attack-based `S-RPD` models for different attackers.

| Dataset | Source Attack | Target Attack | detection accuracy(%) Ori. (%) | Trans.(%) | Diff.(%) | defense accuracy(%) Ori. (%) | Trans.(%) | Diff.(%) | restored accuracy(%) Ori. (%) | Trans.(%) | Diff.(%) |
|---|---|---|---|---|---|---|---|---|---|---|---|
| SST2 | BAE | PWWS | 61.56 | 77.68 | 16.12 | 60.08 | 77.51 | 17.43 | 72.93 | 80.62 | 7.69 |
| | | TEXTFOOLER | | 80.82 | 80.82 | | 78.70 | 78.70 | | 79.13 | 6.20 |
| | PWWS | BAE | 86.59 | 23.89 | −62.70 | 86.45 | 23.70 | −62.75 | 84.68 | 50.69 | −33.99 |
| | | TEXTFOOLER | | 78.82 | 78.82 | | 77.60 | 77.60 | | 76.88 | −7.80 |
| | TEXTFOOLER | BAE | 87.59 | 29.70 | −57.89 | 85.42 | 29.00 | −56.42 | 85.67 | 57.00 | −28.67 |
| | | PWWS | | 85.80 | 85.80 | | 85.71 | 85.71 | | 86.60 | 0.93 |
| Amazon Polarity | BAE | PWWS | 93.76 | 92.69 | −1.07 | 93.76 | 92.61 | −1.15 | 92.85 | 90.95 | −1.90 |
| | | TEXTFOOLER | | 87.25 | 87.25 | | 87.25 | 87.25 | | 86.90 | −5.95 |
| | PWWS | BAE | 96.25 | 50.00 | −46.25 | 96.25 | 50.00 | −46.25 | 94.65 | 83.55 | −11.10 |
| | | TEXTFOOLER | | 92.14 | 92.14 | | 92.14 | 92.14 | | 91.75 | −2.90 |
| | TEXTFOOLER | BAE | 96.49 | 50.57 | −45.92 | 96.49 | 50.57 | −45.92 | 91.95 | 83.15 | −8.80 |
| | | PWWS | | 94.10 | 94.10 | | 94.10 | 94.10 | | 92.30 | 0.35 |
| AGNews | BAE | PWWS | 87.14 | 91.40 | 4.26 | 83.81 | 89.74 | 5.93 | 83.90 | 83.85 | −0.05 |
| | | TEXTFOOLER | | 63.05 | 63.05 | | 62.13 | 62.13 | | 62.30 | −21.60 |
| | PWWS | BAE | 96.72 | 67.69 | −29.03 | 95.33 | 64.47 | −30.86 | 88.80 | 80.90 | −7.90 |
| | | TEXTFOOLER | | 87.90 | 87.90 | | 85.71 | 85.71 | | 80.45 | −8.35 |
| | TEXTFOOLER | BAE | 94.60 | 62.15 | −32.45 | 91.63 | 59.51 | −32.12 | 85.40 | 80.95 | −4.45 |
| | | PWWS | | 97.33 | 97.33 | | 96.23 | 96.23 | | 89.55 | 4.15 |

## RQ4: DOES DETACHED ADVERSARIAL TRAINING OBJECTIVE WORK IN RPD?

To alleviate the performance sacrifice caused by adversarial training on clean data, we adopt the detached adversarial training objective. To verify its feasibility, we employ traditional adversarial training in RPD. The results in Table 9 show that traditional adversarial training works for perturbation defocusing, while the performance drop on clean data is inevitable. We also evaluate ablated RPD without adversarial training objective; the experimental results show that the detection accuracy and restored accuracy increases by $\approx 1\% - 2\%$, this is because the adversarial detection

objective attracts more attention while $\beta = 0$. However, restored accuracy drops $\approx 2\% - 3\%$. Therefore, we believe that detached adversarial training is effective in RPD.

Table 9: The experimental results of RPD based on ensemble adversarial training objective.

| Dataset | Target Model | Clean Acc.(%) | Attacker | Attacked Acc.(%) | Detection Acc.(%) | Detection Diff.(%) | Defense Acc.(%) | Defense Diff.(%) | Restored Acc.(%) | Restored Diff.(%) |
|---|---|---|---|---|---|---|---|---|---|---|
| SST2 | DEBERTA | 94.73 | BAE | 45.96 | 70.97 | 0.42 | 68.28 | −0.79 | 77.00 | −0.70 |
| | | | PWWS | 29.82 | 95.72 | 0.63 | 95.65 | 0.79 | 92.04 | 0.83 |
| | | | TEXTFOOLER | 22.02 | 92.67 | −1.66 | 90.33 | −2.41 | 86.99 | −2.14 |
| Amazon Polarity | DEBERTA | 96.20 | BAE | 56.65 | 82.86 | −3.36 | 82.86 | −3.36 | 89.33 | −2.22 |
| | | | PWWS | 19.40 | 98.32 | 2.07 | 98.32 | 2.07 | 93.20 | −1.45 |
| | | | TEXTFOOLER | 20.80 | 95.40 | −0.26 | 95.40 | −0.26 | 91.12 | −1.53 |
| AGNews | DEBERTA | 92.12 | BAE | 81.80 | 80.95 | −6.71 | 80.95 | −4.20 | 89.06 | −0.09 |
| | | | PWWS | 56.55 | 96.15 | −1.15 | 96.15 | 0.26 | 88.94 | −2.01 |
| | | | TEXTFOOLER | 32.95 | 90.64 | −2.63 | 90.15 | −1.22 | 88.00 | −0.40 |

## A.6 DEPLOYMENT DEMO

We deploy an anonymous demonstration of RPD on Huggingface Space[9], and we provide two examples of this demonstration in Figure 7 to show the usage of RPD. In this demonstration, the user may either enter a new phrase with a label or randomly choose an example from the dataset supplied in order to execute an attack, adversarial detection, and adversarial repair.

---

[9]https://huggingface.co/spaces/anonymous8/RPD-Demo

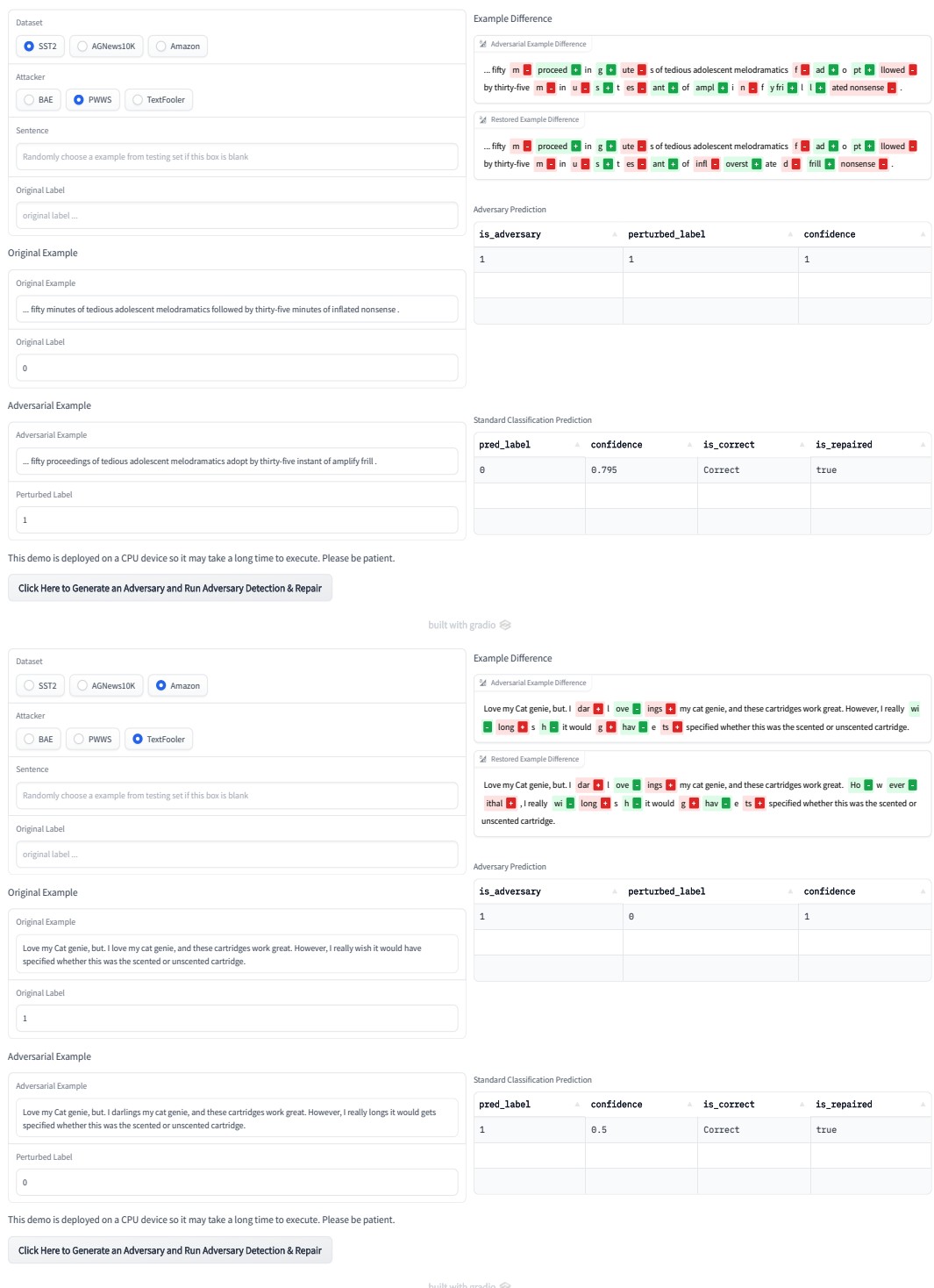

Figure 7: The demo snapshots of adversary detection and defense built on RPD for defending against multi-attacks.

