# OpenReview forum: "Perturbation Defocusing for Adversarial Defense"
_ICLR.cc/2023/Conference — Submitted to ICLR 2023_

### Official Review · Reviewer_6CCu · 2022-10-21

**Confidence:** 3
**Correctness:** 3
**Technical Novelty And Significance:** 3
**Empirical Novelty And Significance:** 3
**Recommendation:** 5

**Clarity, Quality, Novelty And Reproducibility:**

Clarity issues:
* References are not clickable links.
* The acronym “PLM” is undefined, I presume it means “pre-trained language model”.
* I am a bit lost on Section 3, which can be updated to make it easier to read.
	* Section 3.1.2 defines $A_i$ as an attacker for adversary sampling, but it is not clear what “adversary sampling” means. Is it the same as “text adversarial attack” in Section 3.1.1?
	* Section 3.1.2 mentions “$\phi$ indicates the sub-label is neglected in cross-entropy loss calculation”, it is not clear through what mechanism the label is neglected, as it is not defined in text/math.
	* Section 3.2 eq. (6): $\mathcal{A}$ is not previously defined.
	* Section 3.4 mentions “$\mathcal{A}_{PD}$ is an adversarial attacker performing perturbation defocusing”, but the mechanism of perturbation defocusing is not clearly defined in text/math.
* It is not clear through writing how detection affects classification accuracies.  What happens when the resulting model detects an adversarial example during classification?

Question related to reproducibility: Figure 7: the demo adversarial examples appear to be grammatically incorrect with strange changes in semantics. Such examples should be easy to detect.

Novelty: The idea of perturbation defocusing appears sufficiently novel with relatively small accuracy degradations.



**Strength And Weaknesses:**

### Strengths
Perturbation defocusing appears to be a simple, novel and effective technique to restore task accuracies for PLM models under adversarial attacks.

### Weaknesses
* The paper proposes an adversary detection objective. Would it work well under adaptive attacks, i.e. what happens when a white-box attacker can also try to fool your detector?
* Detached adversarial training transfers adversarial examples from a surrogate model for the adversarial training of RPD models, and mentions Table 9 for its effectiveness. However, it is not clear how transferred and online examples may impact model robustness. In addition the caption of Table 9 mentions “ensemble adversarial training objective”, but there is no such objective in the paper.


**Summary Of The Paper:**

This paper propose perturbation defocusing to apply perturbations to a single word in each text input of PLM models to repair potential adversarial inputs. By combining the technique with detection and detached adversarial training, the result show promising task accuracies under adversarial attacks.


**Summary Of The Review:**

The idea of the perturbation defocusing is interesting, but appear to be limited as it removes information from the text, and would thus ultimately impact task performance on clean inputs.

---

### Official Review · Reviewer_Kc7s · 2022-10-21

**Confidence:** 3
**Clarity, Quality, Novelty And Reproducibility:** The paper is not clear at all, it is …
**Correctness:** 2
**Technical Novelty And Significance:** 2
**Empirical Novelty And Significance:** 2
**Recommendation:** 1

**Strength And Weaknesses:**

**Strenghts.**

+ the intuition of injecting noise to counter adversarial attacks is interesting.

**Weaknesses.**

+ **The paper is not readable.** The content of the paper is not clear, as it lacks many technical details that help the reader understand the contribution. In particular:
    + the paper does not give any definition of PLM (I had to infer it from abstract and introduction), and it does not have any background section describing how such technology is developed and trained. Moreover, there are details (like the term *pool*) inside the text that are never explained properly. Such lacking of proper introduction makes the contribution very difficult to understand.
    + the paper does not provide technical understanding on how the correction is applied, but it is left vague. The only technical description is given by the algorithm, which is not informative enough.
    + the detection of adversarial example is an hard problem (as also specified by the authors) but the detection mechanism described by the authors is not clear. The authors should discuss in greater detail how they tackle the problem, since the current discussion is not helping the reader in understanding the concept.
    + no limitations of the approach are discussed.
+ **Adaptive attacks are not discussed.** The authors show that not-adaptive attacks are not able to bypass the proposed mechanism, however the authors did not try to implement an attacker that is aware of such correction mechanism.


**Summary Of The Paper:**

The authors propose a methodology for detecting and correcting adversarial examples computed against text pertained language models (PLM). The correction is done by adding non-toxic words that cancels the presence of adversarial noise inside the input sample.
To show the effectiveness of the methodology, the authors test their framework against language models fine-tuned on specific datasets.

**Summary Of The Review:**

The paper is unclear and very difficult to read at the point that the contribution is not understandable.

---

### Official Review · Reviewer_pngk · 2022-10-26

**Confidence:** 5
**Correctness:** 3
**Technical Novelty And Significance:** 3
**Empirical Novelty And Significance:** 3
**Recommendation:** 5

**Clarity, Quality, Novelty And Reproducibility:**

The novelty of this work lies in the perturbation defocusing method, which is not described in the method section.

I have some questions for the method section:
1. In equation 4, in order to obtain the prediction probabilities, we need to use linear transformation to convert the pooled representation vector into a logic value, but the linear layer is missing in this equation.
2. In equation 5 and 7, they are normal cross-entropy loss function but a standard cross-entropy loss function should be E(y_i log(p_i)), where y_i is the label and p_i is the prediction probability. I think you have reversed the two symbols.
3. What is the difference between the so-called "detached adversarial training objective" and a normal adversarial training objective?

I have some questions for the experiments and results section:
1. What does it mean by "RAT predicts adversaries using an adversarial classifier and predicts natural examples using a standard classifier." ?
2. Could you give a mathematical definition of Defense Accuracy? What is the difference between defense accuracy and restored accuracy?

**Strength And Weaknesses:**

Strength:
1. The proposed perturbation defocusing method can restore the attacked accuracy from very low accuracy to almost the same as the original clean accuracy, which is impressive.

Weaknesses:
1. The proposed RDP method consists of two phases. The first phrase is a normal multi-task learning by combining standard classification objective, the adversarial training, and the adversarial detection, which is not that novel. The second phase is the reactive perturbation defocusing, which is novel. However, this so-called perturbation defocusing method is not described at all. Specifically, the equation 9 is critical, but how to implement the A_{PD} is not described at all. I tried to understand this so-called PD method in the Introduction but failed.
2. The main text only compares with one baseline, RAT (comparison with another two baselines are put in the supplementary, which is not formal). There is no reference to this method. And according to the method description in this work, I know it is an adversarial classifier. But then how do you use this classifier to defend against the adversarial attack? If we only know whether a sample is an adversary or not, then how do we know what is the correct prediction for those detected adversaries?
3. The related work has mentioned that there are three types of defense methods, but this work has compared with one baseline in the main text.

**Summary Of The Paper:**

This work has proposed a new adversarial defense method based on multi-task learning and reactive perturbation defocusing. This method has achieved very impressive restored accuracy.

**Summary Of The Review:**

I am impressed by the good results of this work and would really like to know the details of this method. So I am looking forward to author response to clarify the core method (perturbation defocusing).

---

### Decision · Program_Chairs · 2023-01-20

**Decision:**

Reject

**Justification For Why Not Higher Score:**

Paper not ready for publication.

**Justification For Why Not Lower Score:**

N/A

**Metareview: Summary, Strengths And Weaknesses:**

All reviewers argue for rejection and raised several issues/questions. The authors did not use the rebuttal to clarify these points, so this is a clear rejection.